# A comprehensive search for calcium binding sites critical for TMEM16A calcium-activated chloride channel activity

Jason Tien[1†], Christian J Peters[1†], Xiu Ming Wong[2‡], Tong Cheng[1,3,4], Yuh Nung Jan[1,3,4], Lily Yeh Jan[1,3,4]*, Huanghe Yang[1,3,4]*

[1]Department of Physiology, University of California, San Francisco, San Francisco, United States; [2]Graduate Program in Chemistry and Chemical Biology, University of California, San Francisco, San Francisco, United States; [3]Department of Biochemistry and Biophysics, University of California, San Francisco, San Francisco, United States; [4]Howard Hughes Medical Institute, University of California, San Fransisco, San Fransisco, United States

*For correspondence: Lily.Jan@ucsf.edu (LYJ); huanghe.yang@ucsf.edu (HY)

[†]These authors contributed equally to this work

Present address: [‡]Institute of Molecular and Cell Biology, Singapore, Singapore

**Competing interests:** The authors declare that no competing interests exist.

**Abstract** TMEM16A forms calcium-activated chloride channels (CaCCs) that regulate physiological processes such as the secretions of airway epithelia and exocrine glands, the contraction of smooth muscles, and the excitability of neurons. Notwithstanding intense interest in the mechanism behind TMEM16A-CaCC calcium-dependent gating, comprehensive surveys to identify and characterize potential calcium sensors of this channel are still lacking. By aligning distantly related calcium-activated ion channels in the TMEM16 family and conducting systematic mutagenesis of all conserved acidic residues thought to be exposed to the cytoplasm, we identify four acidic amino acids as putative calcium-binding residues. Alterations of the charge, polarity, and size of amino acid side chains at these sites alter the ability of different divalent cations to activate the channel. Furthermore, TMEM16A mutant channels containing double cysteine substitutions at these residues are sensitive to the redox potential of the internal solution, providing evidence for their physical proximity and solvent accessibility.

## Introduction

Transmembrane proteins of the TMEM16 family include a number of calcium-dependent ion channels, such as the TMEM16A (*Caputo et al., 2008*; *Schroeder et al., 2008*; *Yang et al., 2008b*) and TMEM16B (*Schroeder et al., 2008*; *Stöhr et al., 2009*) calcium-activated chloride channels (CaCCs) and the TMEM16F (*Yang et al., 2012*) small conductance calcium-activated nonselective cation (SCAN) channel. These proteins have been implicated in a wide range of physiological activities (*Hartzell et al., 2009*; *Kunzelmann et al., 2011*; *Berg et al., 2012*; *Scudieri et al., 2012*; *Huang et al., 2012a*) including the control of fluid secretion in epithelia (*Romanenko et al., 2010*; *Huang et al., 2012b*), the regulation of membrane excitability in neurons (*Billig et al., 2011*; *Huang et al., 2012c*), the scrambling of membrane lipids in platelets (*Suzuki et al., 2010*; *Yang et al., 2012*), and the metastasis of certain cancers (*Katoh, 2003*; *Duvvuri et al., 2012*). Despite their important physiological roles, the mechanism by which calcium activates these proteins is still extensively debated.

Based on studies of calcium-activated potassium channels ($K_{Ca}$s) (*Maylie et al., 2004*; *Salkoff et al., 2006*), two models have been proposed to account for the calcium sensitivity of TMEM16A-CaCC (*Figure 1A*). In one model resembling that of the small-conductance $K_{Ca}$ (SK) channels (*Xia et al., 1998*; *Schumacher et al., 2001*), calcium ions interact with calmodulin (CaM), and the physical association of calmodulin with TMEM16A is required to transduce the free energy of ion–protein

**eLife digest** Every cell in the body is surrounded by a barrier called the cell membrane. There are, however, a number of ways that molecules can pass through this membrane to either enter or leave the cell.

Calcium-activated channels are a group of proteins that are embedded within the cell membrane and that allow different ions to pass through the membrane. These proteins are involved in a number of processes in a variety of tissues, for example in the gut, lungs and nervous system. A family of proteins called TMEM16 includes a number of calcium-activated channels that have been recently identified. However, it is not clear how these TMEM16 channel proteins detect the calcium ions that cause them to open.

Two ideas have been suggested: the calcium ions might be detected by a protein called calmodulin, which then forces the channel to open; alternatively, the calcium ions might be detected by the channel protein itself. Tien, Peters et al. have now tested both of these ideas by focusing on a calcium-activated channel protein called TMEM16A, which allows chloride ions to pass through membranes.

The possible role of calmodulin was tested in several ways, such as by preventing it from binding to the TMEM16A protein or from binding to calcium. However, none of these changes affected the opening of the channel; so Tien, Peters et al. concluded that calmodulin is not involved in these channels being activated by calcium ions.

Next, Tien, Peters et al. tested specific parts of the TMEM16A channel protein itself to see if they were involved in calcium detection instead. Proteins are made from smaller building blocks called amino acids, and it is known that some amino acids are more likely to bind to calcium ions than others. There are 38 of these amino acids in the TMEM16A channel that are also found in other members of the TMEM16 family in both fruit flies and mammals. Tien, Peters et al. found that replacing five of these with other amino acids made the channel less sensitive to calcium. Further experiments suggested that four of these five amino acids are clustered at the site where a calcium ion might bind to the TMEM16A channel protein, which suggests that the protein itself can detect calcium directly.

The next challenge will be to understand how calcium ions binding to the site on the TMEM16A channel protein can cause the channel to open to allow the chloride ions to pass through.

interaction into movements of the channel gate (*Tian et al., 2011*; *Vocke et al., 2013*). In a second model similar to that of the large-conductance $K_{Ca}$ (BK) channels (*Schreiber and Salkoff, 1997*; *Shi et al., 2002*; *Xia et al., 2002*; *Wu et al., 2010*; *Yuan et al., 2010*), calcium binds directly to TMEM16A, and this interaction drives channel activation (*Yu et al., 2012, 2014*; *Terashima et al., 2013*).

In this study, we performed a series of experiments designed to clarify the mechanism by which TMEM16A-CaCCs are gated by calcium. First, with multiple lines of evidence, we confirmed that CaM is not required for TMEM16A-CaCC activation. Then, we performed a comprehensive alanine muta-genesis screen of all highly conserved acidic residues in TMEM16A that are predicted to be accessible to the cytoplasm. Based on our measurements of these mutants' apparent calcium sensitivity in inside-out excised patches, we identify four acidic residues (including two acidic residues that were recently implicated in the calcium-dependent activation of TMEM16A-CaCCs [*Yu et al., 2012*]) that satisfy several criteria characteristic of calcium-binding sites. Substitution of each of these residues with various amino acids shifts the apparent sensitivity of TMEM16A-CaCC for different divalent cations, and cysteine residues at these sites can be cross-linked by oxidizing intracellular solutions. Our results thus demonstrate that direct binding of calcium to TMEM16A triggers channel activation independently of calmodulin, identify novel interaction sites between calcium ions and TMEM16A, and lay the groundwork for future studies examining the mechanism of calcium-dependent TMEM16 channel activation.

## Results

### Calmodulin is not required for calcium-dependent TMEM16A-CaCC activation

We began our study of TMEM16A-CaCC's calcium-dependent activation by testing whether CaM is required for channel activity. Reasoning that apparent calcium sensitivity is the most direct measurement

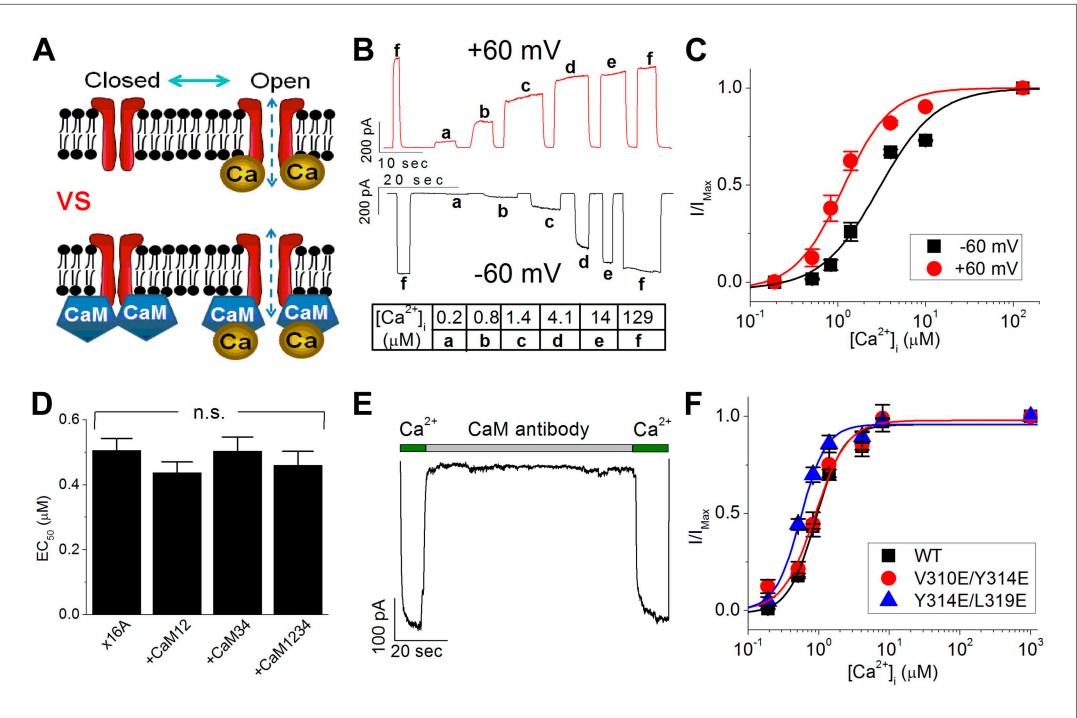

**Figure 1**. Calmodulin (CaM) is not responsible for the calcium-dependent activation of TMEM16A calcium-activated chloride channels (CaCC). (**A**) Two competing models to explain TMEM16A calcium sensitivity have been proposed. It is unclear whether calcium directly binds to TMEM16A-CaCCs (upper panel) or whether CaM is required to mediate the calcium sensitivity of the channel (lower panel). (**B**) Representative current traces of wildtype mouse TMEM16A-CaCC (mTMEM16A) recorded at +60 mV and −60 mV in response to various intracellular calcium concentrations using an inside-out patch clamp configuration. Table indicates the concentration of calcium used. (**C**) Calcium dose–response of mTMEM16A channel at +60 mV and −60 mV. The smooth curves represent fits to the Hill equation ('Materials and methods'). (**D**) Loss-of-function CaM mutants (CaM12, CaM34, CaM1234) did not reduce the apparent calcium sensitivity of the endogenous TMEM16A (x16A) channel in *Xenopus* oocytes. n.s.: non-significant. (**E**) Application of monoclonal anti-CaM antibody CaM85 (2 µg/ml) to the cytosolic face of inside-out patches had no effect on the calcium sensitivity of endogenous *Xenopus* TMEM16A-CaCC. (**F**) Mutating residues reported by *Vocke et al. (2013)* to be in the CaM binding domain of mTMEM16A did not affect apparent TMEM16A-CaCC calcium sensitivity.

The following figure supplements are available for figure 1:

**Figure supplement 1**. Calmodulin (CaM) is not involved in the calcium-dependent activation of TMEM16A-CaCC.

of TMEM16A function, we recorded CaCC currents from GFP-tagged TMEM16A channels at fixed test potentials using an inside-out excised patch configuration. CaCC current amplitudes increased in a calcium-dependent manner until channels reach maximum open probability in our recordings (*Figure 1B*). By fitting the dose–response curve to the Hill equation (*Figure 1C*), apparent calcium sensitivity was derived as the half maximal effective concentration ($EC_{50}$) of calcium required for channel activation. With rapid application of internal solutions, this patch clamp protocol effectively minimized the effect of channel desensitization on our analysis of TMEM16A-CaCC apparent calcium sensitivity (*Figure 1B*). In addition, deriving the apparent calcium sensitivity from individual inside-out patches allowed us to control for variations in protein expression that complicate the interpretation of data derived from other electrophysiology configurations such as whole-cell patch clamp.

Since overexpression of mutant CaM molecules whose EF hand calcium-binding motifs are destroyed has a dominant negative effect on the ability of SK channels to detect changes in intracellular calcium (*Xia et al., 1998*), we tested the effects of these CaM mutations on TMEM16A-CaCCs. We found that co-expression of mutant CaM molecules without functional calcium-binding sites in the N-terminal lobe (CaM12), C-terminal lobe (CaM34), or both lobes (CaM1234) did not reduce the

apparent calcium sensitivity of the endogenous TMEM16A-CaCC in *Xenopus* oocytes (*Figure 1D*), in agreement with a recent study (*Yu et al., 2014*).

Because the failure of CaM mutants to reduce TMEM16A calcium sensitivity could conceivably be due to the inability of the mutant CaM molecules to sufficiently displace endogenous CaM from the putative TMEM16A-CaM complexes, we further clarified the role of CaM in TMEM16A-CaCC gating by conducting four additional experiments to manipulate the potential interaction between CaM and TMEM16A. First, we tried to sequester endogenous CaM molecules by directly applying anti-CaM antibodies to the cytoplasmic side of inside-out patches from *Xenopus* oocytes. In contrast to its potent inhibitory effect on CaM-dependent TRPV1 tachyphylaxis (*Lishko et al., 2007*), the CaM-antibody had no effect on TMEM16A-CaCC currents (*Figure 1E*). Second, we treated endogenous *Xenopus* TMEM16A channels with W7, a potent CaM antagonist that can prevent CaM from binding to its targets. We found that neither acute nor sustained applications of this drug disrupted the calcium sensitivity of TMEM16A channels (*Figure 1—figure supplement 1A,B*). Third, consistent with previous reports (*Xiao et al., 2011*; *Yuan et al., 2013*; *Ni et al., 2014*), we found that barium, a divalent cation that cannot bind to CaM (*Chao et al., 1984*), can still potently activate TMEM16A channels in a dose-dependent manner (*Figure 1—figure supplement 1C,D*). Fourth, mutating three residues (V310, Y314, and L319) that were recently reported to be necessary for CaM-TMEM16A interactions (*Vocke et al., 2013*) did not reduce the apparent calcium sensitivity of the mouse TMEM16A channel expressed in HEK 293 cells (*Figure 1F*). Similar to the results reported by *Vocke et al. (2013)*, the current amplitudes of these mutant channels were much smaller compared to wildtype channels in our inside-out patch recordings (data not shown). However, the absence of any rightward shift in the dose–response curves suggests that these residues do not contribute toward TMEM16A-CaCC calcium sensitivity.

Because we were unable to alter TMEM16A-CaCC activity by inhibiting CaM function, could activate TMEM16A-CaCCs with barium, and could not shift channel $EC_{50}$ by mutating the putative CaM-TMEM16A binding interface, we conclude that CaM is not responsible for the calcium sensitivity of TMEM16A-CaCCs.

## Alanine mutagenesis screen identifies five acidic residues that are important for the calcium-dependent activation of TMEM16A channels

After excluding CaM as the calcium sensor for TMEM16A-CaCCs, we hypothesized that calcium might directly bind to TMEM16A itself. A recent study by *Yu et al. (2012)* found that mutations altering the charge at E698 and E701 (numbered E702 and E705 in the splice variant of TMEM16A used in their experiments) reduced the amplitude of whole-cell currents and altered channel activation and deactivation kinetics, suggesting that a ligand responsible for channel activation may interact with these sites. To test whether the apparent calcium sensitivities of E698Q and E701Q mutant channels are reduced, we measured their steady-state $EC_{50}$ of calcium-dependent channel activation (*Figure 2A,B*). We found that their $EC_{50}$s were shifted to much higher calcium concentrations compared to the $EC_{50}$ of wildtype TMEM16A channels, suggesting that these two acidic residues might be important for a direct interaction between calcium ions and TMEM16A-CaCC.

In proteins, calcium ions tend to bind to the carboxylate moieties of acidic amino acid residues (*Pidcock and Moore, 2001*) and typically require six to eight coordinates (*Dudev and Lim, 2003*). For a comprehensive survey of putative calcium binding residues in TMEM16A, we identified 38 acidic residues (including E698 and E701) that are conserved in all known calcium-activated ion channels in the TMEM16 family (including a *Drosophila* calcium-activated chloride channel [*Wong et al., 2013*]) and that are hypothesized to be solvent accessible from the intracellular compartment (*Figure 2C*). We then systematically replaced each of these residues in the mouse TMEM16A channel with alanine and measured their apparent calcium sensitivity using inside-out patches excised from HEK 293 cells (*Figure 3*). We found that alanine substitutions at E698 and E701 greatly reduced the apparent calcium affinity of TMEM16A-CaCC (*Figure 3*), similar to the effects of glutamine substitutions at these residues (*Figure 2A,B*). Moreover, our screen identified three additional acidic residues (E650, E730, and D734) that were critical for the calcium sensitivity of TMEM16A channels (*Figure 3*). Similar to E698A and E701A, alanine mutations of these three acidic residues resulted in at least a 10-fold reduction of apparent calcium sensitivity. Interestingly, these five residues identified by our screen are confined to the region between the fifth and seventh putative transmembrane segments (*Figure 2C*) (*Yu et al., 2012*).

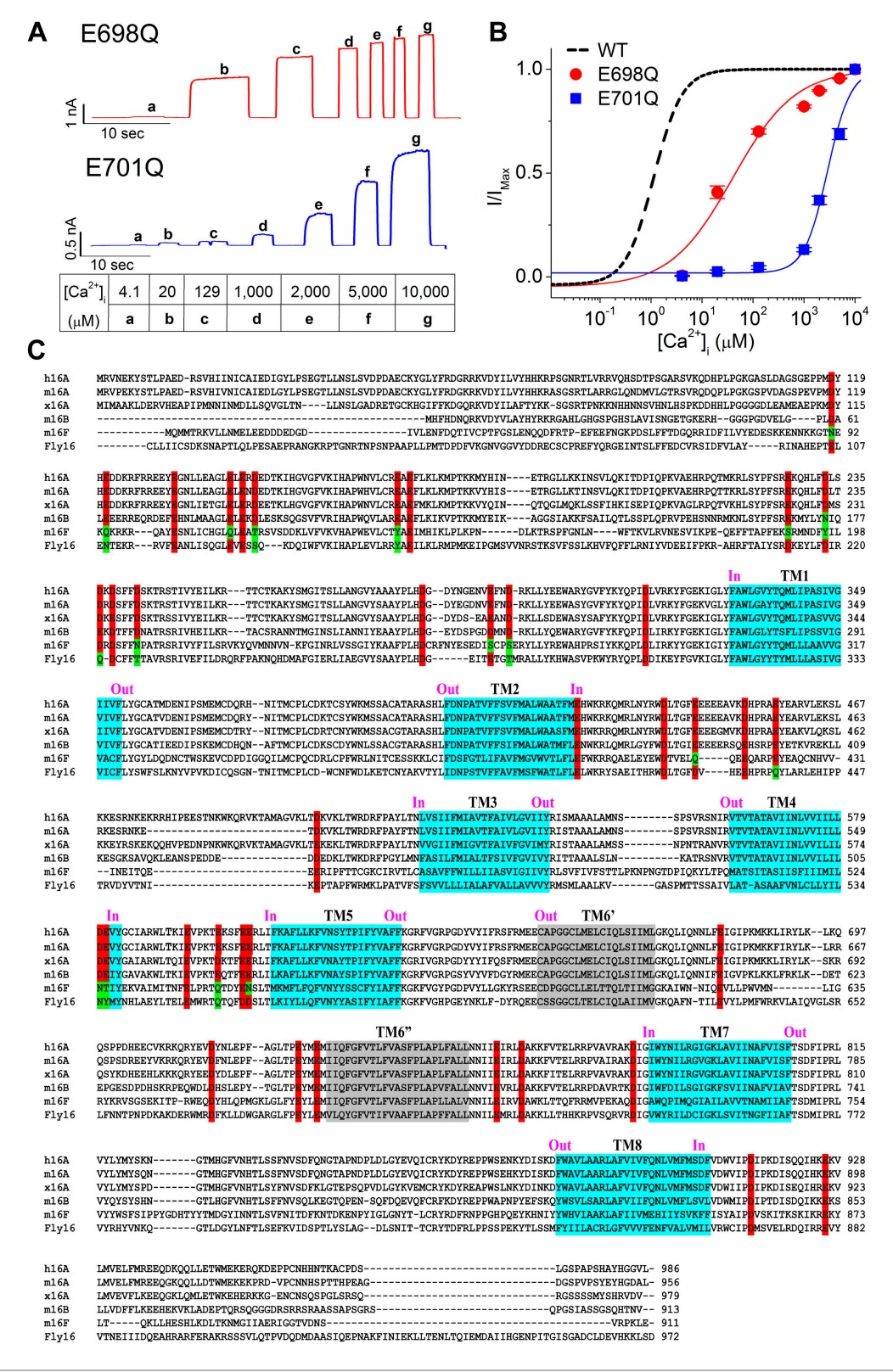

**Figure 2**. Screen for potential calcium-binding residues in TMEM16A-CaCC. (**A** and **B**) Quantification of the apparent calcium sensitivity of E698Q and E701Q (*Yu et al., 2012*) mutant TMEM16A channels. (**A**) Representative current traces of E698Q and E701Q mutants in response to intracellular solutions with different calcium

*Figure 2. Continued on next page*

*Figure 2. Continued*

concentrations recorded at +60 mV. Table indicates the concentration of calcium used. (**B**) Calcium dose–response curves of the mTMEM16A channels at +60 mV. Smooth curves represent fits to the Hill equation. (**C**) Sequence alignment of the calcium-activated TMEM16 channels. h16A, m16A, x16A, m16B, m16F and Fly16 are the human TMEM16A (Uniprot ID #Q5XXA6), mouse TMEM16A (Uniprot ID #Q8BHY3-2), *Xenopus* TMEM16A (Uniprot ID #B5SVV6), mouse TMEM16B (Uniprot ID #Q8CFW1), mouse TMEM16F (Uniprot ID #Q6P9J9) and *Drosophila* TMEM16 channels (Uniprot ID #Q86P24), respectively. Highly conserved acidic residues that are potentially exposed to the cytoplasm are highlighted in red. Some residues with conserved oxygen-containing side chains in m16F and Fly16 are highlighted in green. Putative transmembrane (TM) segments are highlighted in cyan. The controversial TM6 segments are highlighted in gray and labeled as TM6' and TM6", respectively. 'In' and 'Out' indicate the intracellular and extracellular side of the membrane, respectively.

## Altering the side chains of E650, E698, E701, E730, and D734 differentially changes the calcium sensitivity of TMEM16A-CaCCs

Calcium ions interact with different chemical moieties with different affinities depending on the coordination chemistry of the calcium ion (*Sóvágó and Várnagy, 2013*). In order to understand the role of E650, E698, E701, E730, and D734 in TMEM16A-calcium interactions, we replaced the side chains at

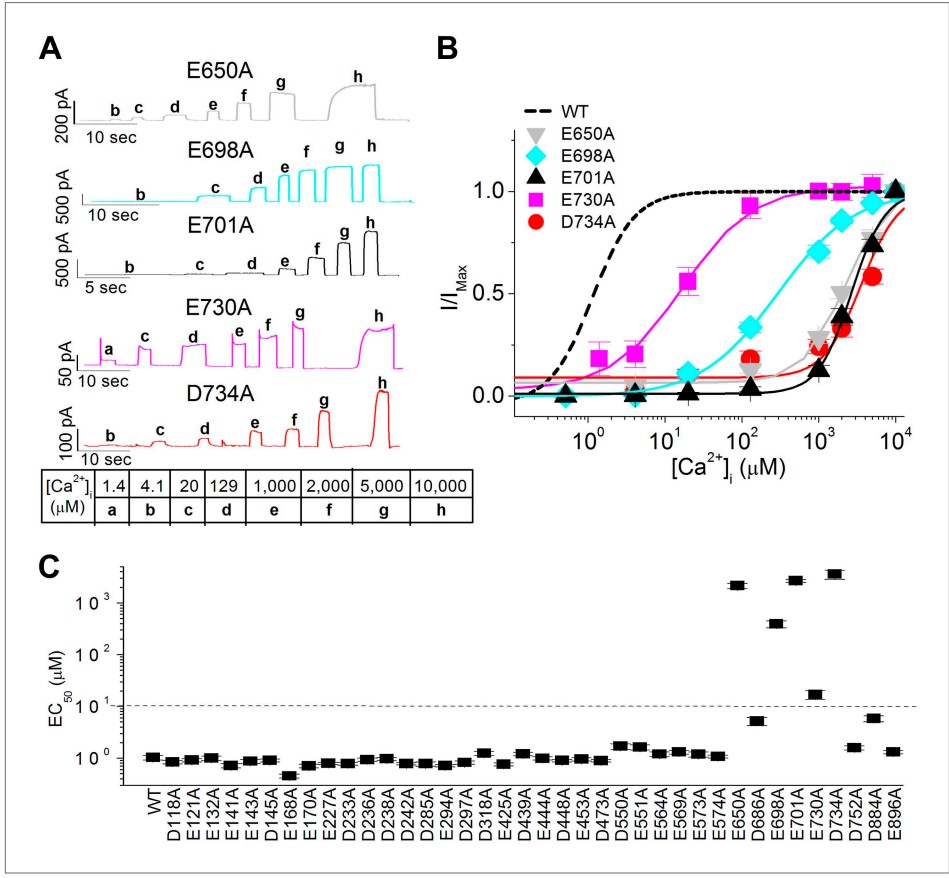

**Figure 3**. Systematic alanine scan of highly conserved intracellular acidic residues identified five mutations that dramatically reduced the apparent calcium sensitivity of TMEM16A-CaCC. (**A**) Representative current traces of the E650A, E698A, E701A, E730A and D734A mutant channels in response to different intracellular calcium solutions recorded at +60 mV. Table indicates the concentration of calcium used. (**B**) Calcium dose–response curves of these mutant mTMEM16A channels at +60 mV. Smooth curves represent fits to the Hill equation. (**C**) Summary of apparent calcium sensitivity (EC$_{50}$s) of all alanine mutants tested. Dotted line indicates a 10-fold increase in EC$_{50}$ compared to wildtype channels.

each of these five potential calcium-binding residues with charge-neutralizing (Cys, Asn, or Gln), charge-reversing (Arg), and charge-conserving (Asp or Glu) groups and examined their effects on the apparent calcium sensitivity of TMEM16A channels (*Figure 4*).

Introduction of cysteine substitutions at these sites replaces the glutamate or aspartate side chain with a small, hydrophobic methyl sulfhydryl group. Understandably, these manipulations had effects comparable to those of alanine substitutions (*Figure 3C*) and reduced the apparent calcium sensitivity of the channel (*Figure 4F*).

Similarly, we reasoned that charge-reversing arginine substitutions of residues that coordinate calcium would have dramatic effects due to the introduction of a positively charged guanidinium group. Indeed, E698R, E701R, E730R and D734R were much less sensitive to internal calcium. Compared to wildtype (WT) channels (EC$_{50}$: 1.0 ± 0.1 µM), the EC$_{50}$s of E698R and E701R were increased by several thousand-fold to 5.0 ± 0.3 mM and 3.3 ± 0.5 mM, respectively, while the effects of E730R were milder, increasing EC$_{50}$ to 0.6 ± 0.3 mM. The most substantial changes were caused by the D734R mutation, which elevated EC$_{50}$ to such a great extent that 10 mM intracellular calcium, the highest concentration of calcium used in this study, was unable to activate the mutant channel. The lack of CaCC current in D734R mutants was not due to defects in protein expression or folding because cadmium ions could still robustly activate the channel (see below). In contrast, the arginine substitution in E650R mutants only produced a small reduction in calcium sensitivity (EC$_{50}$: 6.5 ± 0.8 µM), suggesting that the side chain at position 650 may not be directly involved in calcium coordination.

Because calcium ions tend to be coordinated by oxygen atoms, we also made several mutations to test the role of carboxylate and carbonyl moieties in TMEM16A-CaCC calcium sensitivity. First, we made charge-preserving mutations (Asp or Glu) that alter the length of the amino acid side chain without removing the carboxylate group. If carboxylate moieties are important for TMEM16A-calcium interaction, these mutations may partially preserve TMEM16A-CaCC calcium sensitivity even if the change in side chain length alters the position of the carboxylate group. Second, we made charge-neutralizing mutations (Asn or Gln) that remove the charge of the amino acid side chain without changing its size. If carbonyl groups are involved in calcium coordination, these mutations may maintain the

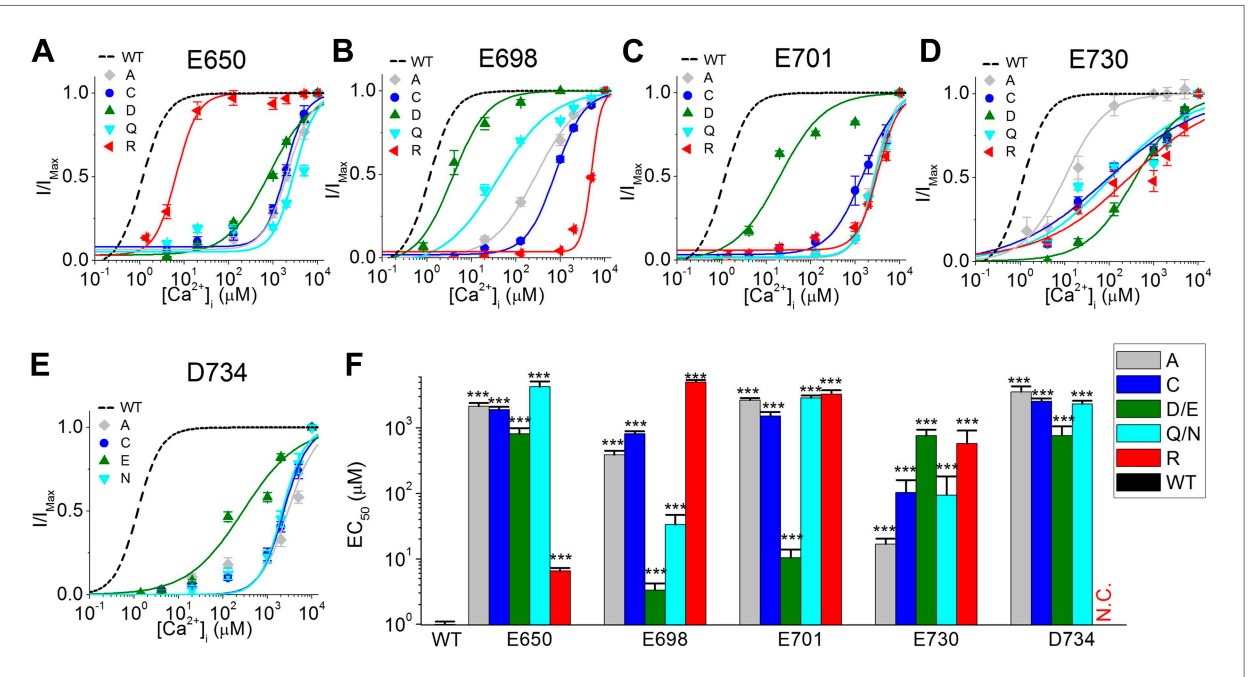

**Figure 4**. The effects of different amino acid side chains on the calcium sensitivity of mutant TMEM16A-CaCC channels indicate that E698, E701, E730 and D734 might be directly involved in binding calcium. (**A–E**) Calcium dose–response curves of (**A**) E650, (**B**) E698, (**C**) E701, (**D**), E730, and (**E**) D734 mutant mTMEM16A channels at +60 mV. Smooth curves represent fits to the Hill equation. Maximum activity was constrained to 1 for these fittings. (**F**) Summary of apparent calcium sensitivity (EC$_{50}$s) of mTMEM16A mutants. N.C.: no obvious CaCC current recorded. ***p<0.001.

orientation of the amino acid residue and preserve the interaction of their carbonyl groups with calcium ions.

We found that the shifts in $EC_{50}$ caused by mutations at E698, E701, and D734 were consistent with these acidic residues playing a direct role in calcium coordination (*Figure 4F*). Charge-preserving mutations in E698D and E701D resulted in only a 3 to 10-fold increase in channel $EC_{50}$ (3.3 ± 0.9 μM and 10.4 ± 3.4 μM, respectively), suggesting that side chain length at these two positions is not critical for calcium binding, while mutations in D734E shifted the $EC_{50}$ by about 800-fold to 0.8 ± 0.3 mM, suggesting that calcium binding may be sensitive to the precise spatial arrangement of the carboxylate group at this position. Although calcium activation was compromised by substituting the acidic residue with a longer side chain, D734E was still more sensitive to calcium than other mutants possessing substitutions at this residue (*Figure 4E*).

In addition, replacing these acidic residues with glutamine or asparagine also increased channel $EC_{50}$, suggesting that the negatively-charged carboxylate group is important for calcium-binding. E698Q mutations resulted in a relatively small decrease in calcium sensitivity ($EC_{50}$: 0.03 ± 0.01 mM) while E701Q and D734N had larger effects ($EC_{50}$: 2.9 ± 0.3 mM and 2.3 ± 0.3 mM, respectively). This data, combined with a dramatic loss of calcium sensitivity in mutants containing arginine, alanine or cysteine mutations at these three sites, supports the idea that side chain carboxylate groups at position E698, E701 and D734 might be directly involved in binding calcium.

Similarly, mutagenesis of E730 suggests that calcium ions may interact with this residue, although not necessarily with its side-chain carboxylate moiety (*Figure 4D,F*). In this case, charge-preserving aspartate mutations increased $EC_{50}$ to 0.8 ± 0.2 mM even though charge-neutralizing glutamine mutations only increased $EC_{50}$ to 0.09 ± 0.08 mM. Since channels bearing mutations that preserve the shape but not the charge of this acidic residue are more sensitive to calcium, it appears that TMEM16A-calcium interactions require the proper conformation of residues at this position, possibly to orient the carbonyl oxygen of the peptide backbone.

The effects of the charge or size-preserving mutations at E650 further support the notion that this residue may not be directly involved in binding calcium. In contrast to the mild effects of a charge-reversing arginine substitution ($EC_{50}$: 6.5 ± 0.8 μM), the charge-preserving aspartate and charge-neutralizing glutamine substitutions increased $EC_{50}$ to 0.8 ± 0.2 mM and 4.3 ± 0.9 mM, respectively, suggesting that neither its carboxylate nor its carbonyl group is involved in calcium coordination. It thus seems likely that E650 might be involved in gating processes downstream of calcium binding while E698, E701, E730, and D734 may interact directly with calcium. This conclusion was reinforced by our study of the effects of E650 mutations on the sensitivity of mutant channels to other divalent cations such as strontium and cadmium (see below).

## Mutations also alter channel sensitivity to other divalent cations

Calcium binding sites often can coordinate other divalent cations that have similar chemical properties or ionic radii similar to those of calcium, such as strontium, barium, and cadmium. Characterizing the selectivity for other divalent cations can provide insight into the composition of a calcium binding site, as shown in previous studies of the cation binding sites involved in the activation of BK channels (*Zeng et al., 2005*; *Zhou et al., 2012*). To further test whether E650, E698, E701, E730, and D734 are involved in divalent cation coordination, we asked whether mutations of these residues also shift the $EC_{50}$ of TMEM16A channel activation by strontium and cadmium ions. Strontium ions can robustly activate TMEM16A channels (*Yuan et al., 2013*; *Ni et al., 2014*) even though their ionic radius (1.13 Å) is larger than that of calcium ions (0.99 Å). Since both strontium and calcium are 'hard' metal ions that prefer coordination with oxygen-containing ligands such as carboxylates and are typically coordinated by six ligands (*Rubin, 1963*; *Dudev and Lim, 2003*), we would expect our various mutations to shift the $EC_{50}$ of strontium and calcium activation in similar directions.

Indeed, we found that the strontium-dependent activity of E698 and E701 mutants were similar to that of their calcium-dependent activity. Compared to the low micromolar strontium sensitivity ($EC_{50}$: 7.7 ± 0.3 μM) of wildtype channels (*Figure 5A,B*), alanine mutations at E698 and E701 increased the $EC_{50}$ of channel activation by strontium to at least the hundred micromolar range ($EC_{50}$: 750 ± 70 μM and 7.6 ± 1.0 mM, respectively). The strontium sensitivity of TMEM16A-CaCCs was drastically reduced by charge-reversing arginine mutations ($EC_{50}$: >10 mM at both sites) and was partially preserved by charge-preserving aspartate or glutamate mutations ($EC_{50}$: 70 ± 20 μM and 90 ± 10 μM, respectively) (*Figure 5D–H*), confirming the importance of side chain charge in divalent cation coordination at these two sites.

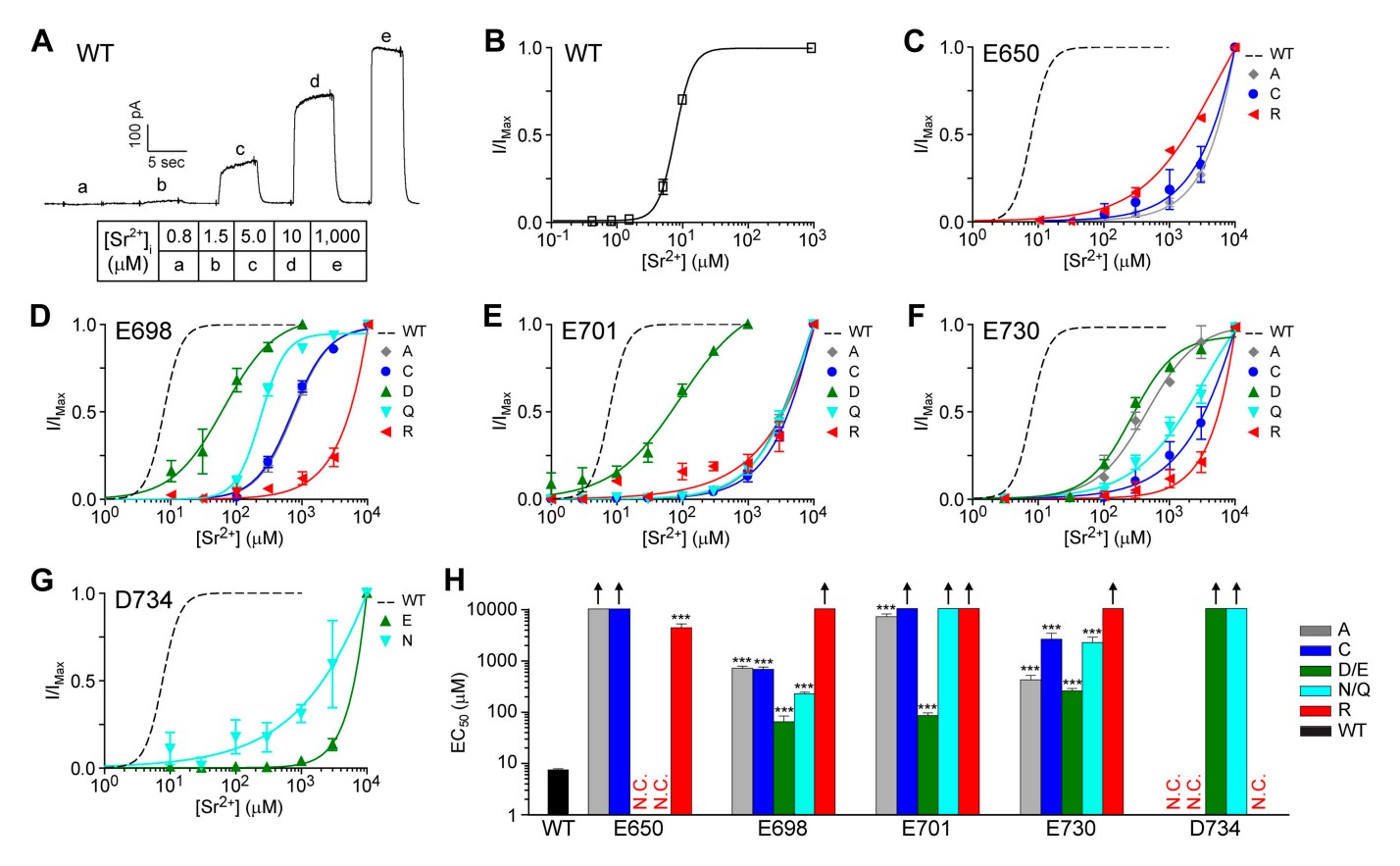

**Figure 5**. TMEM16A channel sensitivity to strontium ions is disrupted by mutations of the identified calcium-binding sites. (**A**) Representative current trace of wildtype mTMEM16A in response to different intracellular strontium solutions recorded at +60 mV. Table indicates the concentration of strontium used. (**B**) Strontium dose–response curve of wildtype mTMEM16A channels at +60 mV. (**C**–**G**) Strontium dose–response curves of the (**C**) E650, (**D**) E698, (**E**) E701, (**F**) E730, and (**G**) D734 mutant mTMEM16A channels at +60 mV. Smooth curves represent fits to the Hill equation. (**H**) Summary of apparent strontium sensitivity (EC$_{50}$s) of mTMEM16A mutants. N.C.: no obvious CaCC current recorded. Upward arrows: estimated strontium EC$_{50}$ >10 mM and cannot be reported with confidence. ***p<0.001.

Similarly, side chain charge appears to be important for strontium-dependent activation of D734 mutants. Although we were not able to accurately estimate the true EC$_{50}$ of any D734 mutants because channel activity was too low to properly normalize current amplitude even at 10 mM strontium, it appears that charge-preserving glutamate mutants are more sensitive to strontium ions than alanine, cysteine, and arginine mutants, which were not activated by strontium at all (***Figure 5H***). The loss of channel activity in these constructs are not due to improper protein folding or export since these mutants can still be robustly activated by intracellular cadmium (***Figure 6G,H***). Consistent with our recordings of D734 mutant channels activated by calcium ions (***Figure 4E***), it appears that although glutamate is able to partially substitute for aspartate at this site, the side chain length as well as the charge of D734 is important for channel activation by hard divalent cations (***Figure 5G***).

Similar to our observations of calcium-dependent channel activation (***Figure 4D***), TMEM16A's apparent EC$_{50}$ for strontium activation seems to depend more on the physical conformation of the side chain than the presence of a carboxylate moiety at residue 730. Channels containing a charge-preserving aspartate mutation at E730 (EC$_{50}$: 0.27 ± 0.03 mM) are not much more sensitive to strontium than channels containing a charge-neutralizing alanine mutation (EC$_{50}$: 0.44 ± 0.10 mM) (***Figure 5F***). Channels containing the aspartate substitution (E730D) had the highest sensitivity for strontium out of the five mutations tested (***Figure 5H***) even though the same mutant channel had the lowest affinity for calcium (***Figure 4F***). It is conceivable that the smaller aspartate side chain at this position can accommodate the larger strontium ion more easily, supporting its direct interaction with these divalent cations.

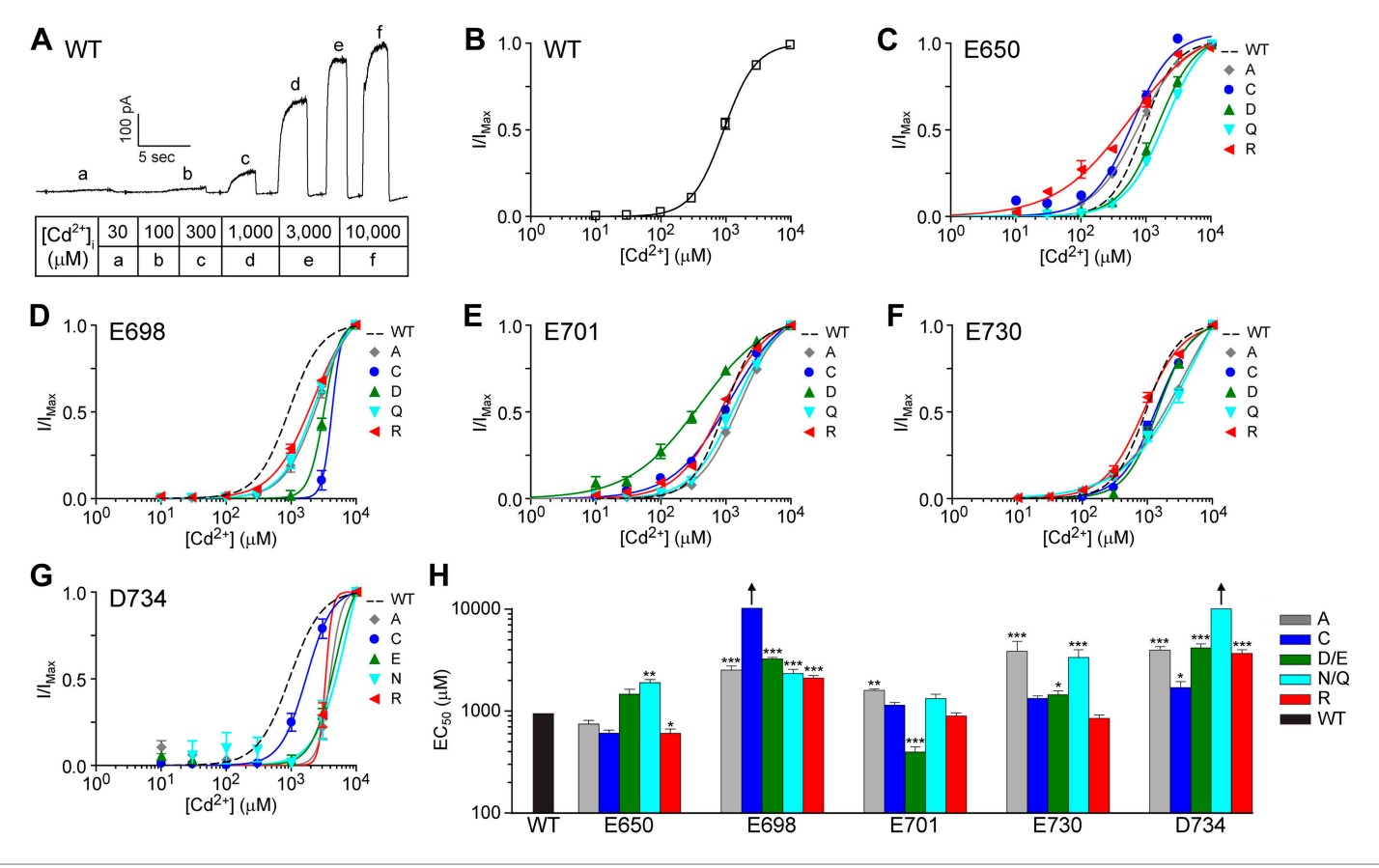

**Figure 6**. TMEM16A channel sensitivity to cadmium ions is disrupted by mutations at the identified calcium-binding sites. (**A**) Representative current trace of wildtype mTMEM16A in response to different intracellular cadmium solutions recorded at +60 mV. Table indicates the concentration of cadmium used. (**B**) Cadmium dose–response curve of wildtype mTMEM16A channels at +60 mV. (**C**–**G**) Cadmium dose–response curves of the (**C**) E650, (**D**) E698, (**E**) E701, (**F**) E730, and (**G**) D734 mutant mTMEM16A channels at +60 mV. Smooth curves represent fits to the Hill equation. (**H**) Summary of apparent cadmium sensitivity (EC$_{50}$s) of mTMEM16A mutants. Upward arrows: estimated cadmium EC$_{50}$ >10 mM and cannot be reported with confidence. *p<0.05; **p<0.01; ***p<0.001.

In contrast to E698, E701, E730, and D734 that display features consistent with their direct interaction with calcium and strontium ions, the effects of the E650 mutations on channel activation by strontium (*Figure 5C*) further support the notion that E650 is not directly involved in metal ion binding, as indicated by their effects on calcium sensitivity (*Figure 4A*). Among all of the channels with mutations at residue 650 from which we could observe strontium-activated currents, the charge-reversing E650R mutant channel exhibited the highest sensitivity to strontium and was the only substitution at that site that yielded an EC$_{50}$ for strontium below 10 mM. In contrast, arginine substitutions of E698 and E730 exhibited the lowest strontium sensitivity as compared to other mutations of these residues, and arginine substitutions of E701 and D734 raised EC$_{50}$ for strontium beyond 10 mM (*Figure 5H*). This is further evidence that E650 probably plays an indirect role in channel activation by divalent cations and that its side chain is likely involved in an allosteric activation mechanism downstream of metal ion binding.

In addition to strontium, we also tested how mutations of these potential calcium binding residues affect the ability of cadmium to activate TMEM16A. Cadmium, with an ionic radius of 0.97 Å that is slightly smaller than that of strontium and calcium, is a divalent cation that is able to substitute for calcium and activate wildtype TMEM16A channels (*Figure 6A,B*). Unlike calcium and strontium ions, however, cadmium is a 'soft' ligand typically coordinated by only four residues (*Dudev et al., 2006*), and it prefers coordination by moieties containing nitrogen and sulfur atoms (*Andersen, 1984*;

*Sóvágó and Várnagy, 2013*). Reasoning that the TMEM16A calcium-binding site is evolutionarily optimized for 'hard' ions like calcium, we hypothesized that the TMEM16A sensitivity for cadmium ions would be low and would not be greatly impacted by mutations that alter the charge of only one acidic residue in the calcium sensor.

Indeed, cadmium ($EC_{50}$: 940 ± 80 μM) is much less effective than calcium ($EC_{50}$: 1.0 ± 0.1 μM) and strontium ($EC_{50}$: 7.7 ± 0.3 μM) at activating the wildtype TMEM16A channel (*Figure 6A,B*). TMEM16A channels were poorly activated by 100 μM cadmium even though it reaches its maximum open probability at 100 μM calcium (*Figure 1C*) and 100 μM strontium (*Figure 5B*). The charge-reversing E650R mutation caused a slight left-shift of the dose response curve of cadmium activation (*Figure 6C*), in contrast to the right-shift observed with the more conservative size-preserving E650Q mutation, further supporting the notion that this residue does not directly coordinate metal ions but rather works downstream of metal ion binding for channel activation. Although mutations of E698, E701, E730, and D734 shifted the $EC_{50}$ of cadmium activation, the differences in $EC_{50}$ between different mutants (*Figure 6D–G*) were much smaller than those observed with calcium and strontium (*Figure 4B–E*, *Figure 5D–G*). It is worth noting that the $EC_{50}$s for cadmium activation of TMEM16A channels containing charge-reversing or charge-neutralizing mutations of these acidic residues were all in the millimolar range, similar to those of wildtype channels and those with charge-preserving mutations (*Figure 6H*), suggesting that the binding of cadmium ions is not as sensitive as the binding of calcium or strontium ions to side chain perturbations of just one of these putative calcium binding residues. This may reflect the difference in coordination chemistry between the soft cadmium ion and the harder calcium and strontium ions.

Since calcium, strontium, and cadmium activation of TMEM16A-CaCC are all affected by mutations of E698, E701, E730, and D734, it appears that all three divalent cations can interact with the same TMEM16A calcium sensor. Because the effects of each mutation are contingent on the divalent cation species used to activate the channel, it is likely that these sites directly alter the metal ion-binding pocket. In addition, because all of our mutations targeted residues on TMEM16A, our results lend further support to the notion that the TMEM16A polypeptide itself encodes the calcium sensor responsible for channel activation.

## E698, E701, E730, and D734 are spatially clustered and exposed to the intracellular solution

Our results implicating E698, E701, E730, and D734 as divalent cation coordinating residues in the TMEM16A calcium-sensing domain suggest that these acidic residues are exposed to the cytoplasm. However, due to the high hydrophobicity of the residues in the region between the putative transmembrane segments 5 and 7, current models of TMEM16A topology (Model A and Model B in *Figure 7C*) do not clearly predict their solvent-accessibility (*Das et al., 2008*; *Yang et al., 2008b*; *Yu et al., 2012*). To confirm that these residues are accessible to internal calcium ions and to demonstrate the spatial proximity of these residues, we tested whether the activity of TMEM16A channels containing double cysteine mutations can be modulated by the redox potential of the internal solution.

In support of a TMEM16A topological model shown as Model C in *Figure 7C* where E698, E701, E730, and D734 are clustered together and facing the cytoplasm, the perfusion of internal solutions containing redox chemicals and 300 μM calcium altered the amplitude of chloride currents through channels containing an E701C/E730C double mutation but not those containing an E701C or an E730C single mutation. Channel activity in oxidizing conditions was significantly smaller than those recorded in reducing conditions, suggesting that formation of a disulfide bridge between E701C and E730C decreases TMEM16A-CaCC activation by calcium (*Figure 7A,B*). Thus, these residues appear to be spatially clustered at a cytoplasm-accessible site.

## Discussion

Calcium is the physiological stimulus that activates calcium-activated chloride channels (CaCCs). Studying how calcium activates the newly discovered TMEM16A-CaCC channel will help us understand its physiological functions. Thus far, the molecular nature of the TMEM16A calcium sensor has been under extensive debate. One theory asserts that a ubiquitous calcium sensing protein, calmodulin (CaM), is responsible for calcium activation of the CaCC (*Tian et al., 2011*; *Vocke et al., 2013*), resembling the mechanism of the small-conductance $K_{Ca}$ (SK) channels. A competing theory asserts

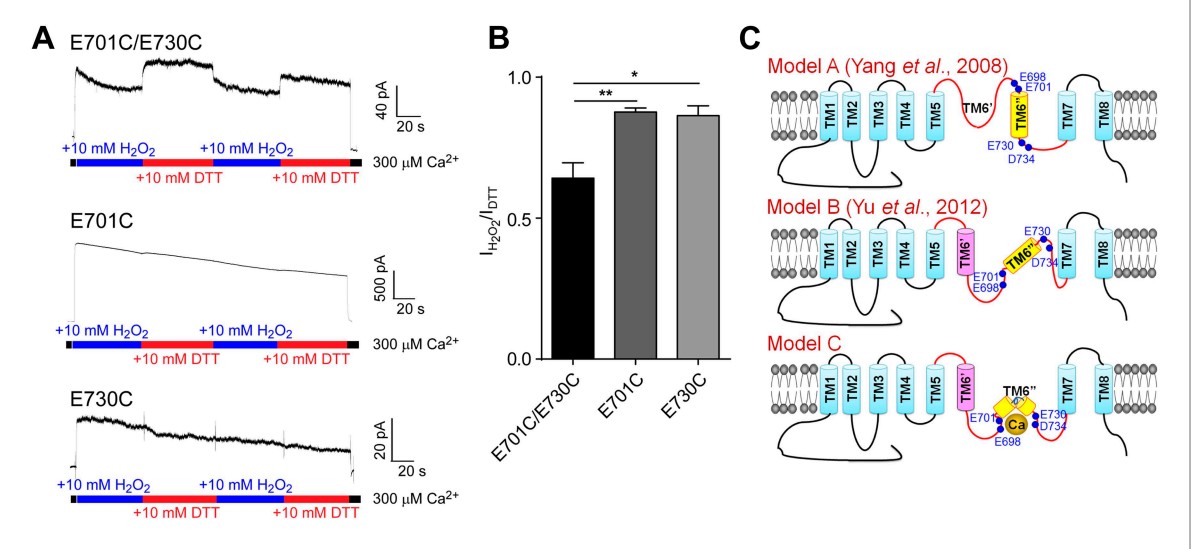

**Figure 7**. Cysteine crosslinking suggests that the calcium-binding residues in TMEM16A-CaCC form a metal ion binding pocket that is exposed to the cytoplasm. (**A**) Representative traces of E701C/E730C, E701C, and E730C mTMEM16A mutants recorded under reducing (DTT) and oxidizing (H₂O₂) conditions. (**B**) Comparison of currents recorded in oxidizing conditions of mutants shown in **A**. When activated for long periods of time, TMEM16A-CaCCs exhibit a persistent decrease in activity, as previously described (*Vocke et al., 2013*). Current amplitudes were measured 60 s after the onset of perfusion and are normalized to currents recorded in reducing conditions. ***p<0.001. (**C**) Schematic illustrating the position of the putative calcium binding residues (E698, E701, E730 and D734) based on two previous membrane topological models (Model A and B) (*Yang et al., 2008b*; *Yu et al., 2012*) and our experimental data (Model C).

that calcium ions can directly bind to the channel itself without the involvement of CaM (*Yu et al., 2012, 2014*; *Terashima et al., 2013*), analogous to the mechanism of the large-conductance $K_{Ca}$ (BK) channels. To settle this debate and to advance the molecular understanding of how calcium activates TMEM16A, we performed a comprehensive screen of evolutionarily conserved intracellular acidic residues in the TMEM16A protein and mapped its calcium sensor to a region between transmembrane segments 5 and 7. By manipulating the charge, size, and reactivity of these side chains, we characterized the mutant channels' sensitivity and selectivity for different divalent cations and showed that four critical acidic residues (E698, E701, E730, and D734) in this region are situated at a solvent-accessible site that likely interacts with calcium in the cytoplasm during channel activation.

In contrast to previous studies of TMEM16A calcium-dependent activation based on whole-cell recordings, we used the inside-out patch clamp to quantify the apparent calcium sensitivity ($EC_{50}$) of the channel in this study. Monitoring channel opening in response to calcium applied to the inside-out patch and comparing channel $EC_{50}$ values is a more direct measurement of calcium-dependent channel activation compared to the methods used in other studies (*Jung et al., 2013*; *Terashima et al., 2013*; *Vocke et al., 2013*). Although recent electrophysiological studies of TMEM16A-CaCCs using the whole-cell patch clamping configuration provided valuable insights toward understanding channel activation (*Yu et al., 2012*; *Jung et al., 2013*; *Vocke et al., 2013*), the difficulty in temporally and spatially controlling intracellular calcium concentrations inherent to the whole-cell patch clamp method renders the quantification of calcium sensitivity difficult. By using inside-out patch recordings to accurately assess channel sensitivity to divalent cations and exploiting differences in the coordination chemistry of hard and soft ligands to deduce the functional role of individual acidic residues, our study based on a mutagenesis screen of acidic residues provides a comprehensive survey of the potential high affinity calcium binding sites within the TMEM16A protein.

Our study is in agreement with the conclusions of *Terashima et al. (2013)* and *Yu et al. (2014)* that calmodulin is not involved in the calcium-dependent activation of TMEM16A channels. Similar to these studies, we were unable to detect alterations of TMEM16A channel activity by manipulating calmodulin function. Moreover, mutations of four putative calcium binding residues E698, E701, E730, and D734 differentially shifted the $EC_{50}$ of channel activation to different calcium concentrations depending on the size and charge of the amino acid side chains, suggesting that these residues are likely

involved in binding calcium. Furthermore, the differences in the way these mutations affect channel sensitivity to the hard calcium and strontium ions and the softer cadmium ions suggest that these mutations directly impact channel–ion interactions rather than acting via an allosteric mechanism. It remains possible that CaM may physically interact with TMEM16A channels under certain conditions as evidenced by previous biochemical studies (*Jung et al., 2013*; *Vocke et al., 2013*). Nevertheless, this interaction, if present, is not responsible for the calcium-dependent activation of TMEM16A-CaCC channels.

It seems likely that calcium directly binds to and activates TMEM16A-CaCC channels in a fashion reminiscent of the calcium activation mechanism of the large conductance, calcium-activated BK potassium channel, one of the most well-characterized calcium-activated ion channels. Both channels contain calcium binding sites that are composed of multiple intracellular acidic residues, and substitution mutagenesis studies of both channels reveal that side chain properties are critical to the binding of calcium and other metal ions such as strontium, barium, and cadmium. Removing carboxylate groups from side chains dramatically reduces or eliminates the ability of both channels to bind calcium, while preserving carboxylate groups with charge-conserving mutations tends to partially rescue calcium sensitivity.

Structural and functional characterizations of BK channels over the past three decades have generated a wealth of information that provides valuable inspiration to the study of TMEM16A-CaCC channels. There are three distinct metal binding sites in the pore-forming subunit of BK channels (*Schreiber and Salkoff, 1997*; *Shi et al., 2002*; *Xia et al., 2002*). Two high affinity binding sites (micromolar) correspond to the calcium bowl in the RCK2 domain (*Schreiber and Salkoff, 1997*) and the acidic residue D367 in the RCK1 domain (*Shi et al., 2002*; *Xia et al., 2002*; *Zhang et al., 2010*), and a low affinity site (millimolar) that can nonspecifically bind various divalent cations corresponds to residues from the membrane spanning domain and the cytosolic RCK1 domain (*Shi et al., 2002*; *Xia et al., 2002*; *Yang et al., 2008a*). These three metal ion binding sites act relatively independently to activate BK channels as mutating any one of them has a minimal impact on the other two sites. Extensive mutagenesis along with characterization of divalent cation selectivity has elucidated the properties of the different metal ion binding sites (*Zeng et al., 2005*; *Zhou et al., 2012*). Micromolar calcium and strontium ions can activate the channel through both the calcium bowl and the RCK1 high affinity sites but not through the low affinity site. Micromolar cadmium only activates the channel via the RCK1 high affinity site; even 300 μM cadmium cannot act on the calcium bowl site to activate the channel. In contrast, 200 μM barium has no effect on the RCK1 site (*Zhou et al., 2012*). Instead, barium ions activate the channel through the calcium bowl site with five times lower affinity than calcium. On the other hand, millimolar concentrations of all these divalent cations can activate the channel through the low affinity metal ion binding site (*Zeng et al., 2005*).

For TMEM16A-CaCC, the apparent calcium ($EC_{50}$: 1.0 ± 0.1 μM) and strontium ($EC_{50}$: 7.7 ± 0.3 μM) sensitivities are similar to that of the high affinity cation binding sites in BK channels. However, the TMEM16A channel is much less sensitive to barium ($EC_{50}$: 330 ± 30 μM) and cadmium ($EC_{50}$: 940 ± 80 μM) than BK channels, which responds to 10 μM barium and cadmium via the calcium bowl and the RCK1 high affinity site, respectively. This difference suggests that the high affinity calcium binding site(s) in the TMEM16A channel might be less effective in binding barium and cadmium ions due to their less optimal coordination chemistry. Alternatively, barium and/or cadmium might activate the TMEM16A channel via an unknown low affinity cation binding site, which remains latent under physiological conditions. This scenario seems plausible as none of our high affinity site mutations abolishes cadmium activation (*Figure 6H*). The identity of this putative low affinity, nonselective cation binding site and its potential involvement in TMEM16A channel activation require further study.

By following up on our comprehensive alanine mutagenesis with multiple point mutations of the putative calcium binding residues and testing their effects on channel activation by different divalent cations, we were able to take advantage of the well-studied chemical properties of divalent cation coordination to analyze these putative calcium binding sites in detail. After these candidate acidic residues were positively identified, conservative mutations preserving either the negative charge (glutamate to aspartate or vice versa) or the approximate side chain volume (glutamate to glutamine, or aspartate to asparagine) were expected to have relatively minor effects on calcium coordination in actual binding sites, considering that either the electrostatic or steric side chain properties would be preserved under those conditions. Conversely, introduction of positively charged arginine side chain would have been expected to produce much more drastic effects. Indeed, this pattern was observed

for four of the five acidic residues identified in the mutagenesis screen. The calcium sensitivity of E698, E701 and D734 was drastically reduced by arginine substitutions, in contrast to the milder effects caused by conservative substitutions. Mutations of E730 followed the same pattern consistent with a divalent ion binding site, though with generally less drastic effects. The E650R mutant, on the other hand, retained the highest sensitivity to calcium of all substitutions at that site, whereas conservative substitutions greatly impacted calcium sensitivity. These results suggest that E650 is unlikely to interact directly with a calcium ion, at least not in the same manner as those of the other four, and we have thus excluded it from our proposed model of the primary calcium interaction site (*Figure 7C*). Given the possibility that E650 may be involved in the transduction of conformational changes associated with calcium binding to trigger the channel's gating machinery, it will be important to further scrutinize its functional role in future studies.

Our analyses using different divalent cations to activate various TMEM16A mutants are also consistent with the notion that E698, E701, E730 and D734, but not E650, are involved in direct interactions with strontium. Strontium is a divalent ion with chemical properties similar to calcium, as it has two electrons in its outer valence shell and is considered a 'hard' ligand (*Rubin, 1963*; *Dudev and Lim, 2003*). Although strontium has a slightly larger ionic radius than calcium, it has been shown previously to robustly activate TMEM16A channels (*Yuan et al., 2013*). The effects of side chain substitutions at those four sites on calcium and strontium sensitivity showed similar trends. In contrast, E650 mutants were generally insensitive to strontium activation, and the mutant displaying the highest sensitivity was again E650R, in keeping with our findings regarding calcium activation.

In addition to strontium, we also employed cadmium as an alternative TMEM16A activator. Cadmium is among the transition metals and acts as a 'soft' ligand, but has an ionic radius nearly identical to that of calcium (*Sóvágó and Várnagy, 2013*). Here, we found that while the wildtype channel was far less sensitive to cadmium ions than calcium ions, the apparent cadmium sensitivities of the mutant channels were insensitive to the side chain manipulations and remain in the submillimolar to millimolar range. In our view, there are at least two plausible explanations for this phenomenon. Firstly, as cadmium ions can be adequately coordinated by fewer negatively charged groups, alterations of just one of the interacting acidic residues may have less drastic effects on cadmium coordination. Alternately, as mentioned previously, it is possible that a second, lower affinity and/or less calcium-selective binding site may exist in the channel that can more readily bind cadmium following mutation in the higher affinity site identified in this study. At present, our results do not allow us to distinguish between these two possibilities, particularly as structural and functional studies of TMEM16A gating are still in their infancy. Nevertheless, the identification of residues forming a high affinity binding site, and our finding that cadmium can activate TMEM16A mutants rendered insensitive to calcium, provide key tools for the future study of channel gating processes downstream of TMEM16A's high affinity binding of divalent cations.

Unlike the high affinity sites of BK channels that are located at the large cytosolic C-terminus (*Salkoff et al., 2006*), the high affinity calcium binding residues (E698, E701, E730, and D734) of the TMEM16A channels are between the putative transmembrane (TM) segments 5 and 7 (*Figure 7C*). This is consistent with recent findings (*Yu et al., 2012*; *Scudieri et al., 2013*) that the third intracellular loop (between TM6' and TM7, *Figure 7C*, Model B) is important for the calcium sensitivity of the TMEM16A and TMEM16B CaCC channels. Previous studies (*Yang et al., 2008b*; *Yu et al., 2012*) have proposed two different models of membrane topology for TMEM16A. The earlier model (Model A in *Figure 7C*) includes a long extracellular or reentrant loop between TM5 and TM6" (*Yang et al., 2008b*), while a subsequent model (Model B in *Figure 7C*) proposes that the peptide between TM5 and TM6" in Model A actually includes one transmembrane segment (TM6') (*Yu et al., 2012*). Our current study provides new insight into the membrane topology of this region by showing that residues 701 and 730 are accessible to the intracellular solvent and are in close proximity such that a disulfide bridge can form between E701C and E730C to impact calcium activation of the channel. This functional study implies that the hydrophobic region spanning 28 residues (including the putative TM6") between E701 and E730 likely forms a short reentrant loop (Model C in *Figure 7C*), bringing together these four acidic residues at both ends of this loop to coordinate calcium ions. Further structural information is needed to validate this model.

In summary, our comprehensive survey of evolutionarily conserved acidic residues has identified several critical residues in the TMEM16A-CaCC that are responsible for its activation by calcium. Our study provides further evidence that the TMEM16A-CaCC channel directly interacts with intracellular

calcium without involving CaM. Interestingly, these putative calcium binding acidic residues are highly conserved among proteins in the TMEM16 family, some of which have been shown to be important in cellular processes ranging from mucus secretion (*Huang et al., 2012b*) to blood coagulation (*Suzuki et al., 2010*; *Yang et al., 2012*) in mammals and host defense in *Drosophila* (*Wong et al., 2013*). The identification of these evolutionarily conserved acidic residues that bind calcium in the TMEM16A-CaCC channel will contribute towards a general understanding of the molecular mechanisms of calcium-activated TMEM16 channels as well as their cellular functions in response to calcium signaling in various cell types and organisms.

## Materials and methods

### Molecular biology and cell culture

cDNAs (Open Biosystems cDNA clones number 30547439, Uniprot identification number Q8BHY3-2) derived from mouse TMEM16A (mTMEM16A) were subcloned into the pEGFP-N1 vector (Clontech, Mountain View, CA) via standard molecular biology techniques. Our clone corresponds to the 'a' splice form identified in *Caputo et al. (2008)* and lacks alternative exons b, c, and d. Site-directed mutations were generated by PCR with Pfu Turbo DNA polymerase following the Quikchange protocol from Agilent. All mutants were verified by sequencing.

HEK 293 cells were cultured in Dulbecco's Modified Eagle Medium (DMEM) supplemented with 4.5 g/l D-glucose, 110 mg/l sodium pyruvate, 584 mg/l L-glutamine, and 10% fetal bovine serum (FBS) and were transfected with Lipofectamine 2000 (Invitrogen, Carlsbad, CA), FuGENE 6 (Promega, Madison, MI), or X-tremeGENE (Roche, Switzerland) and cultured for 1–2 days before recording.

For some experiments testing the effects of calmodulin on channel function, endogenous CaCC currents were recorded from *Xenopus* oocytes without exogenously introducing TMEM16A. Female *Xenopus laevis* were purchased from Nasco (Fort Atkinson, MI). The procedures for harvesting oocytes and housing animals were approved by the UCSF Institutional Animal Care and Use Committee. The loss-of function calmodulin mutants were kindly provided by Dr John Adelman (*Xia et al., 1998*). Defolliculated oocytes were injected with 5–100 ng cRNA and maintained at 17°C in ND96 (96 mM NaCl, 10 mM HEPES, 2 mM KCl, 1 mM $MgCl_2$, pH 7.4) solution for 2–7 days before recording. The monoclonal anti-CaM antibody (CaM85) and the CaM antagonist W7 (*N*-(6-Aminohexyl)-5-chloro-1-naphthalenesulfonamide hydrochloride) were purchased from Invitrogen and Santa Cruz Biotech (Santa Cruz, CA), respectively. For chronic treatments of W7, *Xenopus* oocytes were incubated in ND96 supplemented with 50 µM W7 for 2–6 days before patch recording.

### Electrophysiology

24 hr following transfection, cells were transferred to our standard recording bath solution (described below) for inside-out patch clamp recording. Macroscopic currents were recorded from inside-out patches formed with borosilicate pipettes of 1–5 MΩ resistance. Data were acquired using Axopatch 200-B and Axopatch 700-B patch-clamp amplifiers and pClamp10 software (Molecular Devices, Sunnyvale, CA). All experiments were performed at room temperature (22–24°C).

Unless otherwise stated, all solutions used in this study were based on isotonic 140 mM NaCl. Both the basal extracellular solution and the zero calcium intracellular solution contained 140 mM NaCl, 5 mM EGTA, and 10 mM HEPES. Pipette (extracellular) solutions were supplemented with 1 mM $MgCl_2$. Internal solutions with various calcium concentrations (<100 µM) were prepared with the pH-metric method (*Tsien and Pozzan, 1989*). Briefly, a 'high calcium' solution (140 mM NaCl, 5 mM Ca-EGTA, 10 mM HEPES) and a zero calcium intracellular solution were mixed in different ratios to give various calcium concentrations. The basal internal solution (without calcium buffer) contained 140 mM NaCl, and 10 mM HEPES. For solutions with $[Ca^{2+}]_i \geq 100$ µM, $CaCl_2$ was directly added to the basal internal solutions (without EGTA) and the free $[Ca^{2+}]_i$ was measured with a $Ca^{2+}$-sensitive electrode (Thermo Scientific, Waltham, MA). The pH of all solutions was titrated with N- methyl-D-glucamine (NMDG) or NaOH to 7.2.

To test channel activation by $Sr^{2+}$ or $Cd^{2+}$, inside-out patches were exposed to a series of $Sr^{2+}$ or $Cd^{2+}$-containing solutions prepared by serial dilution of a base solution containing 140 mM NaCl, 10 mM HEPES, and 10 mM $SrCl_2$ or $CdCl_2$ for a range of divalent concentrations from 10 µM to 10 mM. While most constructs required greater than 10 µM of either divalent to be activated, wildtype TMEM16A required solutions with lower $[Sr^{2+}]$. As $Sr^{2+}$ concentration cannot be accurately diluted

below 10 µM, Sr$^{2+}$-EDTA solutions were prepared based on the CaBuf divalent buffering prediction software created by Dr G Droogmans (Department of Physiology, KU Leuven, Leuven, Belgium). For some constructs, 10 mM cation was insufficient to fully activate the channel, and the EC$_{50}$ could not be calculated with confidence and is not reported.

For experiments manipulating the redox potential of the internal solution, 10 mM H2O2 or 10 mM DTT (dithiothreitol) from Sigma (St. Louis, MO) was freshly added (≤1 hr prior to use) to internal solutions containing either 0 µM or 300 µM free calcium.

## Data analysis

Data analysis was performed with Clampfit 10 (Molecular Devices) and Origin 7.5 (OriginLab, Northampton, MA). Concentration dose–response curves were fit to an equation of the form:

$$I / I_{MAX} = \frac{Amp}{1+\left(\frac{K_D}{[C_a]}\right)^H}$$

where $I$ denotes current, $I_{MAX}$ is the maximum current elicited by the highest concentration of divalent cation, $Amp$ is the maximum value of $I/I_{MAX}$ at a given voltage, $K_D$ is the apparent dissociation constant, and $H$ is the Hill coefficient. EC$_{50}$ values were log-transformed for one-way ANOVA and were compared to wildtype values using Tukey's post-hoc test for significance. Values were considered significantly different if $p<0.05$. For some mutant channels whose cation sensitivity was greatly reduced, the current did not reach a plateau even as the cation concentration was raised to 10 mM. In these cases, the EC$_{50}$ values derived from the dose response curves may underestimate the 'true' sensitivity of the mutant channels to these metal ions.

## Acknowledgements

We thank Dr John Adelman for providing the loss-of-function calmodulin mutants and Dr Jianmin Cui for advice on the project. JT is supported by a NIH Ruth L Kirschstein National Research Service Award under grant number 5F31NS076180. CJP is supported by a Junior Personnel Research Fellowship from the Heart & Stroke Foundation of BC and the Yukon. HY is supported by a NIH Pathway to Independence Award (K99NS086916). Research reported in this publication was supported by NIH grant R01NS069229 to LYJ. The content is solely the responsibility of the authors and does not necessarily represent the official views of the National Institutes of Health. YNJ and LYJ are investigators of the Howard Hughes Medical Institute.

## Additional information

### Funding

| Funder | Grant reference number | Author |
| --- | --- | --- |
| National Institutes of Health | 5F31NS076180 | Jason Tien |
| Howard Hughes Medical Institute | | Yuh Nung Jan, Lily Yeh Jan |
| Heart and Stroke Foundation of Canada | | Christian J Peters |
| National Institutes of Health | R01NS069229 | Lily Yeh Jan |
| National Institutes of Health | K99NS086916 | Huanghe Yang |

The funders had no role in study design, data collection and interpretation, or the decision to submit the work for publication.

### Author contributions

JT, Tested the divalent cation effects on these residues, Performed the crosslinking experiment, Analyzed the data, Conducted mutagenesis, Wrote the manuscript; CJP, Tested the divalent cation effects on these residues, Analyzed the data, Wrote the manuscript; XMW, Tested calmodulin effects; TC, Conducted mutagenesis; YNJ, Conceived and designed the study; LYJ, Conceived and designed

the study, Analyzed the data, Wrote the manuscript; HY, Conceived and designed the study, Tested calmodulin effects, Conducted the screen with alanine mutations and identified/verified the key residues for calcium binding, Analyzed the data, Wrote the manuscript

## Ethics

Animal experimentation: This study was performed in strict accordance with the recommendations in the Guide for the Care and Use of Laboratory Animals of the National Institutes of Health. All of the female *Xenopus laevis* were handled according to approved institutional animal care and use committee (IACUC) protocol (#AN086415-03A) of the University of California, San Francisco. The procedures for harvesting oocytes and housing animals were performed in strict accordance with the protocol, and every effort was made to minimize suffering.

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
