## [Decision Letter]

Thank you for sending your work entitled “A comprehensive search for calcium binding sites critical for TMEM16A calcium-activated chloride channel activity” for consideration at *eLife.* Your article has been favorably evaluated by a Senior editor and 3 reviewers, one of whom, Richard Aldrich, is a member of our Board of Reviewing Editors.

The Reviewing editor and the other reviewers discussed their comments before we reached this decision, and the Reviewing editor has assembled the following comments to help you prepare a revised submission.

This is an interesting and clearly written paper that provides new information regarding potential Ca^2+^ binding sites that regulate TMEM16 calcium-dependent Cl^-^ channels. The approach is systematic, focusing solely on acidic residues that are conserved among a variety of TMEM16 isoforms. Some of the new results are somewhat confirmatory. First, results in Figure 1 extend previous work suggesting that CaM does not mediate Ca^2+^-dependent activation. Second, the present results confirm the role of E698 and E701 as potential Ca^2+^-sensing residues. However, particularly in regard to the importance of E698 and E701 (along with other residues), the present results are more extensive and rigorous, since the use of excised inside out patches allows definition of the Ca^2+^ dependence of activation in both WT and mutated constructs.

The authors argue that four of five glutamate/aspartate residues could directly contact divalent cations because of different mutation effects on the channel activation by ‘hard’ and ‘soft’ divalent cations. Finally, the channel with E701C/E730C double mutations is sensitive to oxidation/reduction (redox) while the channels with single mutation (E701C or E730C) are not, arguing that the two introduced cysteine residues may form a disulfide bond and that E701 and E730 are thus solvent accessible and close to each other.

A limitation of the paper is that it seems to be largely descriptive, with little attempt to place the properties of this presumed ‘binding site’ within the context of other known Ca^2+^ binding sites, including ‘non-traditional’ sites. For some other presumed sites, some information regarding selectivity among different divalent cations is available, but this is not discussed. This topic and others that may require additional elaboration are mentioned below.

The results of alanine scanning revealed five acidic residues whose mutations significantly alter the EC50 of calcium activation. The authors exclude E650 as a potential residue in the divalent cation binding pocket, while suggesting E730 and E734 to be in direct contact with the activating divalent cations. The patterns of the mutation effects in various E730 and E734 mutants, like that of the E650 mutations, are not straightforward for making such a conclusion. A comparison of the strontium and cadmium activations in E650 mutants with those of E730 and E734 mutants may help strengthen the conclusion that E730 and E734 are in the cation-binding pocket while E650 is not.

The modification of the E701C/E730C double mutant by oxidation/reduction requires further characterization. Based on Figure 7, it appears that the oxidation and reduction effects are instantaneous, or at least as fast as the current activation by 300 micromolar calcium. If the redox modulation, as suggested by the authors involves a disulfide bond, such fast oxidation and reduction rates seem unusual. Further evidence to strengthen that this redox modulation indeed involves a disulfide bond but is not simply an effect due to oxidation/reduction of both cysteines would be helpful. This clarification is important for the argument that E701 and E730 are spatially clustered.

In constructing dose-response curves, the divalent cation-activated currents are normalized to the current induced by the same divalent cation at 10 mM, which in many cases is not a saturating concentration, as judged from the current induced by the highest two concentrations. It is therefore not certain how accurate the reported EC_50_ values are (particularly in the experiments when cadmium is used as the activating ligand). This issue is important because the authors emphasize the different mutation effects on the EC_50_ values of the ’hard’ and ‘soft’ metal cations.

Although the authors provide concentration-response curves for Ca^2+^ (and other divalents) for all the examined constructs, it seems surprising that no comments were made about the apparent Hill coefficients for these curves. Although Hill coefficients are really only an empirical descriptor that relates to actual molecular events in only complicated ways, they can help guide thinking. In the present case, the concentration-responses exhibit features that seem rather surprising. First, some mutations, particularly those causing partial shifts in affinity, also change the apparent slope (i.e., Hill coefficient). Second, the mutations that cause the greatest loss of affinity seem to have apparent slopes more similar to WT. Third, except for the D734R construct, none of the mutations seem to completely abolish the activation effect of Ca^2+^ (or Sr^2+^ and Cd^2+^).

Some further explanation is required. The change in slope seems difficult to explain in terms of a simple change in affinity of a single site on each subunit. One explanation might be that there are really multiple distinct divalent sensitive sites, and only when the higher-affinity site is weakened, does the second site become revealed. Of course, that would require that the net effect of both sites working together is not additive, in the WT construct. Might the similarity of the concentration-response curves for all of the most severe mutations (except E730?) suggest that some low affinity site unrelated to any of the sites examined is still present? For other potential Ca^2+^ binding site residues in other channels (MthK, BK) or even in CaM, is it the case that when key residues are mutated, Ca^2+^ can still be bound, but only at weaker affinity? It would seem important that these issues be raised and discussed and the results here be contrasted to some of these other cases.

Another omission: no attempt was made to place the ‘selectivity’ of the presumed site within the context of other known Ca^2+^ binding sites. It looks like the apparent affinities for activation by Cd^2+^ and Ba^2+^ are similar, but no comment is made on this. Furthermore, the Cd^2+^ and Ba^2+^ affinities appear not too much different from the sensitivity that persists to Ca^2+^ and Sr^2+^ after the strongest effective mutations at each of the important residues. Might this also suggest a very low affinity, non-selective divalent effect, perhaps not unusual in channels?

---

## [Author Response]

We have addressed the reviewers’ concerns by performing additional experiments (Figures 5, 6 and 7) and substantially revising the text, especially the Discussion section. Point-by-point responses are provided below.

*This is an interesting and clearly written paper that provides new information regarding potential Ca*^*2+*^
*binding sites that regulate TMEM16 calcium-dependent Cl- channels*. *[…]*

*The authors argue that four of five glutamate/aspartate residues could directly contact divalent cations because of different mutation effects on the channel activation by ‘hard’ and ‘soft’ divalent cations. Finally, the channel with E701C/E730C double mutations is sensitive to oxidation/reduction (redox) while the channels with single mutation (E701C or E730C) are not, arguing that the two introduced cysteine residues may form a disulfide bond and that E701 and E730 are thus solvent accessible and close to each other*.

*A limitation of the paper is that it seems to be largely descriptive, with little attempt to place the properties of this presumed ‘binding site’ within the context of other known Ca*^*2+*^*binding sites, including ‘non-traditional’ sites. For some other presumed sites, some information regarding selectivity among different divalent cations is available, but this is not discussed. This topic and others that may require additional elaboration are mentioned below*.

For this new channel family, there is no existing physical model to quantitatively describe the behavior of TMEM16A calcium-activated chloride channel (CaCC) activation. We believe that our efforts to identify and characterize the calcium sensor will lay the groundwork for future studies to understand the molecular activation mechanism.

We now include in the Discussion section a comparison of the putative calcium-binding site in the TMEM16A-CaCC channel with other known calcium-binding sites, especially the non-EF-hand sites in BK channels. Further discussions with regard to selectivity among different divalent cations are also incorporated (see below).

*The results of alanine scanning revealed five acidic residues whose mutations significantly alter the EC50 of calcium activation. The authors exclude E650 as a potential residue in the divalent cation binding pocket, while suggesting E730 and E734 to be in direct contact with the activating divalent cations. The patterns of the mutation effects in various E730 and E734 mutants, like that of the E650 mutations, are not straightforward for making such a conclusion. A comparison of the strontium and cadmium activations in E650 mutants with those of E730 and E734 mutants may help strengthen the conclusion that E730 and E734 are in the cation-binding pocket while E650 is not*.

Strontium and cadmium activation of E650 mutations have been characterized and were added to the revised Figures 5 and 6. Arginine mutations of a calcium binding residue usually will destroy or greatly diminish calcium binding. However, E650R showed the highest sensitivity to strontium and cadmium among all the E650 mutations tested. In fact, E650R is the only mutant at this site that conducts appreciable current with strontium activation. This is consistent with our observation with calcium ions. These experiments thus further support our conclusion that E650 does not directly bind calcium, but rather is critical for gating transduction upon divalent cation binding. We have incorporated the relevant description and discussion in the revised manuscript.

*The modification of the E701C/E730C double mutant by oxidation/reduction requires further characterization. Based on*
Figure 7*, it appears that the oxidation and reduction effects are instantaneous, or at least as fast as the current activation by 300 micromolar calcium. If the redox modulation, as suggested by the authors involves a disulfide bond, such fast oxidation and reduction rates seem unusual. Further evidence to strengthen that this redox modulation indeed involves a disulfide bond but is not simply an effect due to oxidation/reduction of both cysteines would be helpful. This clarification is important for the argument that E701 and E730 are spatially clustered*.

The previous Figure 7 shows recordings from excised patches exposed to oxidizing or reducing solutions interspersed with calcium-free solutions, thereby leaving the impression of instantaneous channel modifications. We have modified the protocol to include 300 μM calcium throughout the recording episode and have recorded for a longer period of time. As shown in the revised Figure 7, the oxidation and reduction rates are not that fast.

It appears that the oxidizing reaction continues on the order of minutes after the application of H_2_O_2_, and DTT is able to quickly reverse the effect caused by disulfide bond formation. Our protocol of holding the channel in high calcium while changing the redox environment appears to show a decrease in channel activity in the reducing condition separate from channel desensitization, although the effects of desensitization become more pronounced the longer we hold the inside-out membrane patch.

We agree with the reviewer that it is important to scrutinize the effects of disulfide formation of the double cysteine mutant. In the absence of structural information for TMEM16A, it is difficult to demonstrate the proximity of these two residues more directly. We believe our strategy of observing channel activity in oxidizing versus reducing conditions is persuasive since the single cysteine substitution controls do not exhibit the same behavior.

*In constructing dose-response curves, the divalent cation-activated currents are normalized to the current induced by the same divalent cation at 10 mM, which in many cases is not a saturating concentration, as judged from the current induced by the highest two concentrations. It is therefore not certain how accurate the reported EC_50_ values are (particularly in the experiments when cadmium is used as the activating ligand). This issue is important because the authors emphasize the different mutation effects on the EC50 values of the ‘hard’ and ‘soft’ metal cations*.

Indeed, the current did not reach a plateau for some mutant channels as the divalent cation concentration was raised to 10 mM. The EC_50_ values for these mutations were underestimated and may not reflect the “true” sensitivity to these metal ions. However, it is technically challenging to apply divalent cations at concentrations higher than 10 mM, which usually causes adverse effects on the quality of the inside-out patches and the current recording. Using V_1/2_ values derived from tail currents following various test potentials is an alternative way to quantify calcium sensitivity. Unfortunately, the TMEM16A-CaCC channel desensitizes quickly over time, rendering it impossible to use this long measurement procedure to quantify the apparent calcium sensitivity.

A description was added to “Data Analysis” in the Method section to alert the readers about this limitation. We believe that this limitation does not invalidate our Conclusions.

*Although the authors provide concentration-response curves for Ca*^*2+*^
*(and other divalents) for all the examined constructs, it seems surprising that no comments were made about the apparent Hill coefficients for these curves. Although Hill coefficients are really only an empirical descriptor that relates to actual molecular events in only complicated ways, they can help guide thinking. In the present case, the concentration-responses exhibit features that seem rather surprising. First, some mutations, particularly those causing partial shifts in affinity, also change the apparent slope (i.e., Hill coefficient). Second, the mutations that cause the greatest loss of affinity seem to have apparent slopes more similar to WT. Third, except for the D734R construct, none of the mutations seem to completely abolish the activation effect of Ca*^*2+*^
*(or Sr*^*2+*^*and Cd*^*2+*^*)*.

As we discussed above, the dose-response curves for some mutant channels could not be accurately fitted with the Hill equation either because the current did not reach a plateau as the divalent cation concentration was raised to 10 mM or because the current amplitude was drastically reduced. Considering the empirical nature of Hill coefficients and the limitation of our experiments, we restrained from relying on this parameter for further mechanistic insights.

The Hill coefficient of calcium dose-response curve for the wildtype channels is about 2, suggesting that each monomer of the dimeric TMEM16A channel might contain at least one high affinity calcium binding site. The dose-response curves for some mutations are shallower (reduced Hill coefficients), suggesting these mutations may partially destroy the high affinity site(s). On the other hand, the mutations that cause the greatest loss of affinity leave the Hill coefficient unaltered, raising the possibility that the high affinity site(s) might be completely destroyed causing some low affinity site(s) to dominate the channel activation. In this scenario, each monomer may contain such a low affinity site to give the Hill coefficient of around 2. As we currently do not have any information regarding the molecular nature of this putative low affinity site(s) in this admittedly speculative scenario based on less-than-reliable estimates of the Hill coefficient, we feel it is better to leave a discussion of the Hill coefficient out of the manuscript.

We have incorporated into the revised Discussion section statements regarding the possibility that the residual sensitivity of channels bearing these potent mutations may derive from a latent low affinity site(s).

*Some further explanation is required. The change in slope seems difficult to explain in terms of a simple change in affinity of a single site on each subunit. One explanation might be that there are really multiple distinct divalent sensitive sites, and only when the higher-affinity site is weakened, does the second site become revealed. Of course, that would require that the net effect of both sites working together is not additive, in the WT construct. Might the similarity of the concentration-response curves for all of the most severe mutations (except E730?) suggest that some low affinity site unrelated to any of the sites examined is still present? For other potential Ca*^*2+*^
*binding site residues in other channels (MthK, BK) or even in CaM, is it the case that when key residues are mutated, Ca*^*2+*^
*can still be bound, but only at weaker affinity? It would seem important that these issues be raised and discussed and the results here be contrasted to some of these other cases*.

Thank you very much for the great suggestion. There is precedent for calcium binding to some of the mutated binding sites with weaker affinity. One example is the mutational effects of D367 in the RCK1 high affinity binding site of the BK channel (Zhang, et al *P.N.A.S*., 2010, 107: 18700–18705). With greatly reduced affinity to calcium, D367E and D367N mutant channels nonetheless exhibited significant calcium sensitivity, in contrast to channels bearing other mutations of the same residue, suggesting that retaining the side chain carboxylate or carbonyl groups still allows coordination of calcium. Similar findings have also been reported for some of the calcium bowl mutations (Bao & Cox, J. G. P., 2004, 123: 475–489). We observed similar side chain specific patterns in the TMEM16A-CaCC channels in this study. The possibility of the existence of an unknown low affinity metal binding site is also discussed and included in the Discussion section.

*Another omission: no attempt was made to place the ‘selectivity’ of the presumed site within the context of other known Ca*^*2+*^
*binding sites. It looks like the apparent affinities for activation by Cd*^*2+*^
*and Ba*^*2+*^
*are similar, but no comment is made on this. Furthermore, the Cd*^*2+*^
*and Ba*^*2+*^
*affinities appear not too much different from the sensitivity that persists to Ca*^*2+*^
*and Sr*^*2+*^
*after the strongest effective mutations at each of the important residues. Might this also suggest a very low affinity, non-selective divalent effect, perhaps not unusual in channels*?

A comparison of the calcium binding sites in TMEM16A and the Slo1 BK channel is now included in the Discussion section. The possibility of the existence of a low affinity, nonselective divalent cation binding site is also discussed.

In addition, we tried to activate several of our mutants with Ba^2+^. Of these, only E730D and E730Q (Figure 8) yielded appreciable current. Thus, it appears that Ba^2+^ does not interact with TMEM16A in the same way as Cd^2+^ does. Rather, like other ’hard’ divalent cations, barium does not tolerate mutations at the calcium-binding sites identified in our study. In contrast, all of the mutant channels tested can still be activated by Cd^2+^.Author response image 1.Barium dose-response curve of wildtype and mutant mTMEM16A channels at +60 mV. E730D and E730Q were the only mutants that produced appreciable activity out of the channels tested.The pattern of bariumdependentactivation is not similar to that of cadmium-dependent activation.